# Impact of Comorbidities of Patients with Psoriasis on Phototherapy Responses

**DOI:** 10.3390/ijms23179508

**Published:** 2022-08-23

**Authors:** Belén Fatás-Lalana, Joaquín Cantón-Sandoval, Lola Rodríguez-Ruiz, Raúl Corbalán-Vélez, Teresa Martínez-Menchón, Ana B. Pérez-Oliva, Victoriano Mulero

**Affiliations:** 1Departamento de Biología Celular e Histología, Facultad de Biología, Universidad de Murcia, 30100 Murcia, Spain; 2Instituto Murciano de Investigación Biosanitaria (IMIB)-Arrixaca, 30120 Murcia, Spain; 3Centro de Investigación Biomédica en Red de Enfermedades Raras (CIBERER), Instituto de Salud Carlos III, 28029 Madrid, Spain; 4Servicio de Dermatología, Hospital Clínico Universitario Virgen de la Arrixaca, 30120 Murcia, Spain

**Keywords:** psoriasis, comorbidities, diabetes, phototherapy, zebrafish

## Abstract

A retrospective study of 200 psoriasis patients and 100 healthy donors in a Spanish cohort was carried out to study the comorbidities associated with psoriasis and their association with the response to phototherapy. The results showed a higher incidence of psychiatric disease, liver disease, kidney disease, hypertension, heart disease, vascular disease, diabetes, gastrointestinal disease, autoimmune and infectious diseases, dyslipidemia, and psoriatic arthritis in patients with psoriasis than in the control group. The incidence of comorbidities was higher in psoriasis patients over 40 years old than in the control individuals of the same age, which could be indicative of premature aging. Phototherapy was seen to be an effective treatment in cases of moderate-severe psoriasis, total whitening being achieved in more than 30% of patients, with women showing a better response than men. Narrow-band ultraviolet B was found to be the most effective type of phototherapy, although achievement of PASI100 was lower in patients with liver disease, hypertension, heart disease, vascular disease, or diabetes. Strikingly, liver disease and anemia comorbidities favored therapeutic failure. Finally, zebrafish and human 3D organotypic models of psoriasis point to the therapeutic benefit of inhibiting the glucose transporter GLUT1 and the major regulator of blood glucose dipeptidyl peptidase 4. Our study reveals that specific comorbidities of psoriasis patients are associated to failure of phototherapy and, therefore, need to be considered when planning treatment for these patients.

## 1. Introduction

Psoriasis is a chronic and inflammatory disease that causes redness, swelling, and peeling of the skin. There are numerous studies about the multiple comorbidities associated with psoriasis [1,2,3], and the conclusions obtained suggest that it should be considered as a systemic disease. Interestingly, a study conducted on patients with psoriasis revealed that they are not aware of the risk of comorbidities associated with the disease [4].

Among the best-known comorbidities are depression, psoriatic arthritis, and inflammatory bowel disease [5]. However, more recent studies have also linked psoriasis to other pathologies, such as metabolic syndrome, cardiovascular diseases, celiac disease, non-alcoholic fatty liver disease, kidney disease, infections, and certain neoplasms [6]. Importantly, psoriasis significantly increases the risk of mortality, especially if severe. This mortality is associated with cardiovascular and non-cardiovascular causes, including liver disease, kidney disease, infection, and neoplasms [7]. More studies are needed to better understand the different pathologies associated with psoriasis and their clinical relevance.

We aim to identify the comorbidities of a cohort of 200 patients with generalized plaque psoriasis, the influence of age and sex, their association with the response to phototherapy, and the relevance of diabetic comorbidity on skin inflammation. We found a higher number of comorbidities in patients with respect to the controls, the incidence depending significantly on gender and age. Moreover, our data indicate that narrow-band ultraviolet B (NB-UVB) is the most effective type of phototherapy for plaque psoriasis, although the effectiveness was greatly affected by the comorbidities. Zebrafish and 3D organotypic models of human psoriasis demonstrated the relevance of diabetic comorbidity in skin inflammation.

## 2. Results and Discussion

### 2.1. Psoriatic Patients Present More Associated Comorbidities Than Healthy Subjects

An analysis of comorbidities was performed in a cohort of 200 psoriatic patients compared with 100 individuals belonging to the control group (Appendix A). All the medical histories from the previous 10 years were revised. The results showed an increased number of comorbidities in psoriatic patients compared with control individuals (2.59 ± 2.16 vs. 1.25 ± 1.36, Figure 1a). In addition, the maximum number of comorbidities observed in the group of patients with psoriasis was 10 compared with 5 in the control group (Figure 1a).

Several studies have shown the relevance of different comorbidities associated with psoriasis [1,3,8]. In the present study, we found that the incidence of 11 of the 16 studied comorbidities showed a statistically significant difference between the psoriatic patients and control individuals (Figure 1b). As regards any psychiatric pathology, patients with psoriasis showed a higher incidence associated with the level of an anxiety-depressive pathology, as previously reported [9,10,11]. Despite the obvious influence of stigmatization of psoriatic patients, some authors with similar results to ours suggest that a potential cause of this anxiety-depressive pathology is the intrinsic inflammatory process of psoriasis, i.e., the production of proinflammatory cytokines [12,13].

Another interesting observation of our study is that liver disease also showed a significantly higher prevalence in psoriatic patients than in control subjects (Figure 1b). This agrees with several studies in the literature that correlate psoriasis with a higher prevalence of non-alcoholic fatty liver disease [14,15]. Similarly, the incidence of hypertension was significantly higher in patients with psoriasis than in the control group (Figure 1b). Hypertension has been shown to be one of the most prevalent comorbidities of psoriasis, and its incidence is associated with a higher body mass index, with the development of psoriatic arthritis, and with a later-onset psoriasis [16].

Our data also revealed a statistically significant higher incidence of both cardiac and vascular pathologies in psoriasis patients than in control subjects (Figure 1b). Although further studies are needed, previous findings seem to correlate cardiac pathologies with the systemic inflammation caused by psoriasis, in which the spleen seems to play a key role [17]. In addition, a higher incidence of acute myocardial infarction in psoriasis patients has been reported [18].

We found a significantly higher incidence of dyslipidemia in psoriasis patients than in the control group (Figure 1b), in agreement with previous studies [19,20,21].

Finally and importantly, the incidence of diabetes showed a statistically significant higher incidence in psoriasis patients than in the control group (Figure 1b), as recently reported by [3]. This correlation appears to be especially important when the onset of psoriasis is late; however, other studies found no such correlation with the patient’s age or with the severity of psoriasis [22].

### 2.2. Men Show More Comorbidities Associated to Psoriasis Than Women

The group of patients with psoriasis presented on average more than double the comorbidities that the control group in both men and women. In addition, the number of men and women who did not present any comorbidities were less than half that in the psoriasis one. The pathologies with a statistically significant higher incidence in the men with psoriasis than in the men of the control group were psychiatric pathologies, hypertension, heart disease, arthritis, and psoriatic arthritis (Figure 2a), whereas the women with psoriasis were more prone to liver disease, diabetes, and psoriatic arthritis (Figure 2b). In this respect, psoriatic arthritis obviously showed a statistically significant difference in all comparisons, due to the absence of this pathology in the control group.

### 2.3. Premature Aging Is Associated with Comorbidities in Psoriatic Patients

With respect to the age variable, the number of comorbidities did not statistically differ between psoriasis patients and control subjects aged 18–39 (Figure 3a). This lack of significance could be due to the small number of samples, since twice as many comorbidities were observed in the group of patients with psoriasis. Similarly, none of the comorbidities showed a statistically significant difference between groups (Figure 4a). In sharp contrast, in both 40–59 and >60 year-old subjects, a significantly higher number of comorbidities was observed in patients with psoriasis than in the control group (Figure 3a): 0.89 ± 1.14 vs. 2.62 ± 1.85 in 40–59 year-old and 2.31 ± 1.32 vs. 4.38 ± 2.39 in >60 year-old subjects. In 40–59-year-old subjects, psychiatric pathologies, hypertension, vascular pathology, diabetes, and autoimmune and infectious diseases showed a statistically significant higher incidence in psoriatic patients than in control individuals (Figure 4b). In >60 year-old subjects, the comorbidities that were significantly more frequent in psoriasis patients than in control individuals were liver disease, kidney disease, hypertension, heart disease, and diabetes (Figure 4c). Although there was no evidence to suggest that psoriasis shortens life expectancy, accelerated aging might be responsible for the higher incidence of comorbidities that we found in psoriasis patients with a lower chronological age. Thus, 40–59-year-old psoriasis patients showed a similar incidence of comorbidities than >60 year-old control subjects. This would agree with the older biological age, assayed by the Horvath clock, observed in female psoriatic patients [23], an observation that would warrant further investigation.

### 2.4. Impact of Comorbidities, Gender and Type of Phototherapy

At the Virgen de la Arrixaca University Clinical Hospital (HCUVA, Murcia, Spain), the largest hospital in south-eastern Spain, several types of phototherapies have been applied during the last 10 years to treat patients with psoriasis, including long-wavelength ultraviolet A (UVA) (42.5%), UVA with the use of skin-photosensitizing furocoumarins (PUVA) (18%), NB-UVB (14.5%), and a combination of UVA and NB-UVB (25%). The most widely used phototherapy was UVA and the least used was NB-UVB. Only 2.5% of patients of the cohort analyzed abandoned the treatment, dropping out due to adverse effects (1.5%) or to lack of response to treatment (1%). Of the five patients who abandoned treatment due to the appearance of adverse effects, three received PUVA, one UVA, and another NB-UVB phototherapy. The two patients who abandoned treatment due to lack of response were receiving UVA. However, it is PUVA that is considered to produce more adverse effects [24,25].

No studies on the different comorbidities in psoriasis and their influence on the treatment of phototherapy could be found. Our study revealed a significantly lower percentage of psoriatic patients with liver disease, hypertension, heart disease, vascular disease, and diabetes comorbidities that ended in complete remission of psoriasis lesions (PASI100) in response to phototherapy (Figure 5a). Noticeably, we found that liver disease and anemia comorbidities favored therapeutic failure, assessed as the failure to reach PASI50 (Figure 5b). The discrepancies observed in the impact of anemia in patients reaching PASI100 and PASI50 might be explained by the low incidence of anemia in our patient cohort (14 out of 200). However, 7 out of 14 psoriatic patients with anemia showed therapeutic failure, i.e., they failed to achieve PASI50. It is important to point out that the sample was homogeneous with respect to initial PASI and, therefore, this does not account for the treatment efficacy. Although further studies are needed, our findings show for the first time that specific comorbidities should be considered when selecting the most favorable therapeutic options for psoriatic patients.

As regards the effect of gender, women showed a better response to phototherapy treatment and reached a higher percentage of all PASIs (PASI50, PASI75, PASI90, and PASI100) than men (Figure 5c). No studies have previously objectively analyzed the impact of gender on the efficacy of phototherapy. Nevertheless, a study based on a subjective assessment by the patients reported that women were more dissatisfied with the outcome of the treatment than men [26]. Taken together, our data show that male psoriatic patients suffered a greater number of comorbidities and, in turn, had a worse response to phototherapy treatment.

To assess the efficacy of the different types of phototherapies administered, a comparison was made of the PASIs obtained and the type of phototherapy applied. A lower percentage of patients receiving UVA achieved complete remission of the disease (PASI100) than patients treated with other types of phototherapies, whereas a higher percentage failed to respond, assessed as reaching at least PASI50 (Figure 5d). In contrast, patients treated with NB-UVB phototherapy showed greater therapeutic success in all PASIs (PASI50, PASI75, PASI90, and PASI100) compared to the rest of the phototherapies administered (Figure 5e). Although these results indicate that NB-UVB phototherapy should be the phototherapy of choice and replace UVA phototherapy, which is routinely used, the efficacy of the different phototherapies is controversial. Although one study found PUVA to be more efficacious than NB-UVB, especially in refractory psoriatic lesions [24], another showed a better and longer-lasting response of patients to NB-UVB [27]. In addition, it has also been reported that the combination of UVA and NB-UVB had a similar efficacy to monotherapy with NB-UVB [28].

### 2.5. Zebrafish Larvae and Human 3D Psoriatic Skin Organoids Are Valid Models to Study Comorbidities Associated with Psoriasis

As diabetic comorbidity has been found to critically affect the efficacy of phototherapy, and a recent study identified a therapeutic target for psoriasis based on the different requirements of keratinocytes for glucose in healthy and inflamed skin [29], we next sought to determine the impact of glucose levels and Dipeptidyl Peptidase 4 (DPP4) inhibition, a widely used treatment in diabetic patients, in preclinical models of psoriasis.

We used a zebrafish deficient in the serine protease inhibitor, Kunitz type 1a (Spint1a), which shows keratinocyte hyperproliferation and cell death, loss of epithelial integrity, neutrophilia, and increased infiltration of neutrophils in the skin (Figure 6a) [30,31]. Pharmacological inactivation of Glut1 transporter by increasing doses of the drug WZB117 reduces skin neutrophil infiltration (mobilized from the caudal hematopoietic tissue, CHT) and keratinocyte aggregates, surrogates of keratinocyte hyperproliferation (Figure 6b). The ability of exogenous galactose and glucose to restore skin neutrophil infiltration and keratinocyte aggregates in Spint1a-deficient larvae treated with WZB117 confirmed the essential role of carbohydrate metabolism in fueling skin inflammation (Figure 6c). As nicotinamide phosphoribosyltransferase (NAMPT)-derived NAD^+^ hyperproduction in inflamed skin is required to sustain keratinocyte death and inflammation [30], we next analyzed the effects of NAD^+^ in combination with the pharmacological inhibition of Glut1. The results showed that WZB117 failed to alleviate skin inflammation (assayed as skin neutrophil infiltration and the number of keratinocyte aggregates) in the presence of exogenous NAD^+^ (Figure 6d), suggesting that glucose metabolism sustains high levels of NAD^+^ that, in turn, promote keratinocyte death and inflammation.

As several studies have found that treatment of psoriatic patients with either DPP4 inhibitors or incretin mimetics not only ameliorate type 2 diabetes mellitus (T2DM) but psoriasis [32,33,34], we evaluated the effect of the genetic inhibition of Dpp4. The results showed that Dpp4 inhibition alleviated neutrophil skin infiltration and the number of keratinocyte aggregates (Figure 7a). In addition, the beneficial impact of Dpp4 inhibition was fully reversed by exogenous glucose, confirming that Dpp4 regulates skin inflammation in Spint1a-deficient zebrafish larvae through glycemic control (Figure 7b).

These results prompted us to evaluate the impact of glucose metabolism in human organotypic 3D skin models of psoriasis (Figure 8a). The results showed that the pharmacological inhibition of either GLUT1 (Figure 8b) or DPP4 (Figure 8c) reduced the mRNA levels of pathology-associated genes, including the pro-inflammatory *S100A8*, *DEFB4*, and *NAMPT*, and the proliferation *PCNA* biomarkers. These results not only confirm previous observations showing that keratinocytes are more dependent on glucose for their growth in inflamed skin [29], but also reveal the cell-autonomous effects of DPP4 in keratinocytes. This latter observation is not surprising, since it has been reported that DPP4 plays an important role in the differentiation of the epidermis and its cornification [35]. In addition, it has also been found that the inhibition of DPP4 with sitagliptin to control blood glucose in a patient with psoriasis and T2DM, alleviated psoriasis [36], and a randomized controlled trial has recently demonstrated that sitagliptin combined with NB-UVB significantly improved psoriasis compared with NB-UVB alone in patients with moderate psoriasis without T2DM [37]. In conclusion, the human organotypic 3D skin models of psoriasis showed the same results as in zebrafish and is also a valid model for studying the influence of diabetic comorbidity in psoriasis.

## 3. Materials and Methods

### 3.1. Study Design and Patient Cohort

A retrospective study of patients from the Dermatology Service of the Virgen de la Arrixaca University Hospital (HCUVA, Murcia, Spain), recruited between 2019 and 2021, was carried out. The clinical records of the patients were analyzed without any prior screening. A total of 300 individuals participated in the study (Appendix A), 100 of whom were part of the control group and whose data and medical records were provided by the Health Center of Arrabal (Zaragoza, Spain). The 200 patients included in the study group had received phototherapy for the last 10 years at the HCUVA. In addition to the specialist reports, the electronic prescription, the most recent blood analyses, and the phototherapy reports with initial and final PASI assessment were included.

The only characteristics that permitted inclusion in the control group were: older than 18 years, a recent blood analysis (less than 12 months), and not having psoriasis. The pathologies of the patients in the control group were studied through the reports of medical specialists sent to the primary care physician, as well as through their pharmacological treatment included in their electronic prescriptions. 

### 3.2. Analysis of Comorbidities

Between January 2019 and March 2021, a review of the medical reports (physical and computerized) of all the cases was carried out, with a subsequent review on 17 April 2021 after completing the analysis. With the collected data, a database in Excel format was generated, in which the following variables were taken into account: year of birth, age group (18–39, 40–59 and >60 years old), gender (male/female), presence and number of comorbidities, which included osteoporosis, psychiatric pathologies (subdivided into anxious-depressive pathology and other psychiatric pathologies), liver disease, renal disease, hypertension, cardiac pathology, vascular pathology, diabetes, gastrointestinal pathology, dermatological diseases, autoimmune and infectious diseases, hyperuricemia, dyslipidemia, arthritis, psoriatic arthritis, and anemia. For the study of comorbidities, the reports of the specialists and the pharmacological treatment for each patient, as assessed from their electronic prescriptions, were reviewed.

### 3.3. Analysis of Phototherapy Treatments

For the evaluation of the type of phototherapy applied to each patient, the following parameters were considered: previous phototherapy treatment (yes/no), number of phototherapy cycles, cycle start date and end date, current phototherapy treatment, starting, increase and final dose, previous and current cumulative doses, frequency of sessions, current cumulative dose, initial and final PASI, and improvement with phototherapy (PASI50, PASI75, PASI90, PASI100).

### 3.4. Animals

Wild-type zebrafish (*Danio rerio* H.) lines AB, TL, and WIK obtained from the Zebrafish International Resource Center (ZIRC) were used and handled according to the zebrafish handbook [38]. The zebrafish lines *Tg(lyz:dsRED2)^nz50^* [39] and *spint1a^hi2217Tg/hi2217Tg^* [40] were described previously. 

### 3.5. Genetic Inhibition in Zebrafish

The CRISPR RNA (crRNA) obtained from Integrated DNA technologies (IDT) with the following target sequence were used: *dpp4* crRNA: 5′-CACGATGTCCAGTACTTAGC-3. The guide was resuspended in duplex buffer at 100 μM, and 1 μL was incubated with 1 μL of 100 μM trans-activating CRISPR RNA (tracrRNA) at 95 °C for 5 min and then 5 min at room temperature to form the complex. One microliter of this complex was mixed with 0.25 μL of recombinant Cas9 (10 mg/mL) and 3.75 μL of duplex buffer [30]. The efficiency of each crRNA was determined by the TIDE webtool (https://tide.nki.nl/, accessed on 14 March 2022) [41]. Crispant larvae, 3 days post fertilization were used in all the studies.

### 3.6. Chemical Treatments in Zebrafish

Zebrafish embryos were manually dechorionated at 24 hpf. Larvae were treated from 24 hpf to 48 hpf or 72 hpf by bath immersion at 28 °C. Incubation was carried out in 6- or 24-well plates containing 20–25 larvae/well in egg water (including 60 μg/mL sea salts in distilled water) supplemented with 1% dimethyl sulfoxide (DMSO) [30]. The inhibitors and metabolites used and the concentrations tested and their targets are shown in Appendix A.

### 3.7. Imaging of Zebrafish Larvae

Live imaging of 72 hpf larvae was obtained using buffered tricaine (200 μg/mL) dissolved in egg water [30]. Images were captured with an epifluorescence LEICA MZ16FA stereomicroscope with green and red fluorescent filters. All images were acquired with the integrated camera on the stereomicroscope and were analyzed to determine number of neutrophils in the larvae and their distribution.

### 3.8. Human Organotypic 3D Models

Insert transwells (Sigma-Aldrich MCHT12H48) were seeded with human foreskin keratinocytes (Ker-CT, ATCC CRL-4048) on the transwells in 300 μL CnT-PR medium (CellnTec) in a 12-well format. After 48 h, cultures were switched to CnTPR-3D medium (CELLnTEC) for 24 h and then cultured at the air-liquid interface for 17 days [30]. From day 12 to 17 of the air-liquid interphase culture, the Th17 cytokines IL17A (30 ng/mL) and IL22 (30 ng/mL) were added. Pharmacological treatments consisting of WZB117 at 100 μM, DPP4 at 100 μM, and all-trans retinoic acid (ATRA) at 1 μM (Appendix A), were applied from day 14 to 17. The culture medium was refreshed every two days. On day 17, the tissues were harvested for gene expression analysis.

### 3.9. Statistical Analysis

The statistical program SPPS PASW Statistics for Mac was used. Both Microsoft Excel and the above program were used to make the graphs. A descriptive statistical analysis was performed for both groups, including the distribution percentage according to sex and age, the mean age and standard deviation, the presence, absence, and number of the different comorbidities, and the percentage of both with respect to their respective study groups. In patients with psoriasis, the distribution of the type of phototherapy received was analyzed using descriptive statistics. In addition, the achievement or lack thereof of PASI 50, 75, 90, and 100 was also assessed.

Data were analyzed by analysis of variance (ANOVA) and a Tukey multiple range test to determine the differences between groups with Gaussian data distribution (square-root transformation was used for percentage data). Non-parametric data were analyzed by Kruskal–Wallis test and Dunn’s multiple comparisons test. The differences between two samples were analyzed by Student *t*-test. The contingency graphs were analyzed by means of the Chi-square (and Fisher’s exact) test.

## 4. Conclusions

We observed a higher incidence of several comorbidities in psoriasis patients over 40 years old than in the control individuals of the same age, which could be indicative of premature aging. Phototherapy was seen to be an effective treatment in cases of moderate-severe psoriasis, with women showing a better response than men. NB-UVB was found to be the most effective type of phototherapy, although achievement of PASI100 was lower in patients with liver disease, hypertension, heart disease, vascular disease, or diabetes. Strikingly, liver disease and anemia comorbidities were found to favor therapeutic failure. Finally, zebrafish and human 3D organotypic models of psoriasis reveal the relevance of diabetic comorbidity in skin inflammation and suggest a therapeutic benefit of inhibiting GLUT1 and DPP4. These results indicate that comorbidities need to be considered when planning treatment for patients with psoriasis.

## Figures and Tables

**Figure 1 ijms-23-09508-f001:**
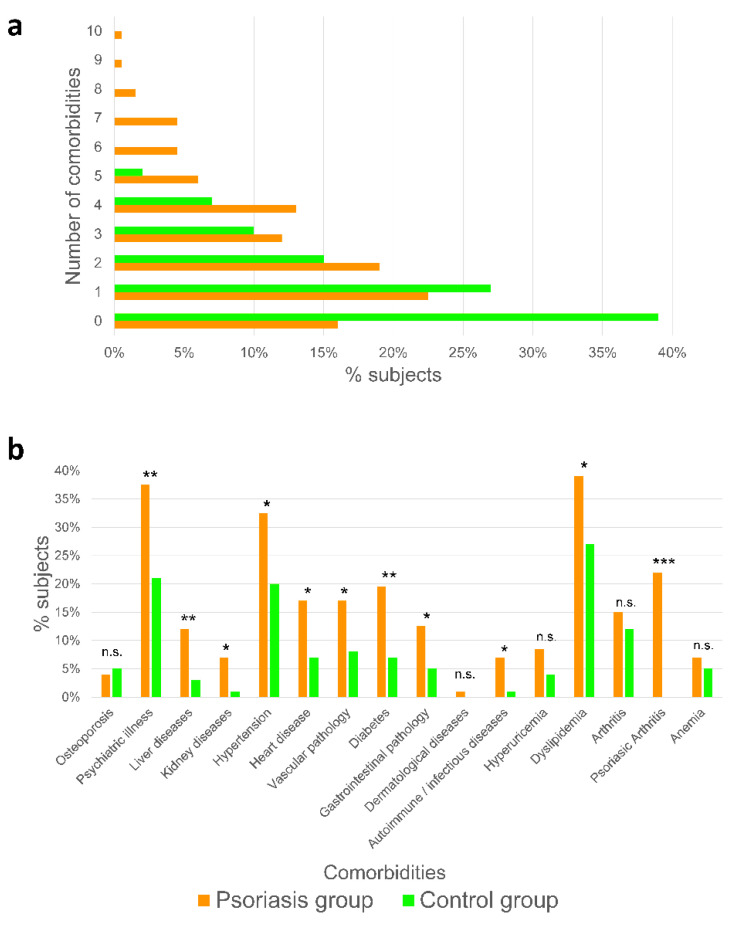
Psoriasis patients have more comorbidities than control individuals. Number of comorbidities (**a**) and percentage of each comorbidity (**b**) in psoriasis and control group. *p*-Values were calculated using a Chi-squared test. * *p* < 0.05; ** *p* < 0.01; *** *p* < 0.001; n.s., non-significant.

**Figure 2 ijms-23-09508-f002:**
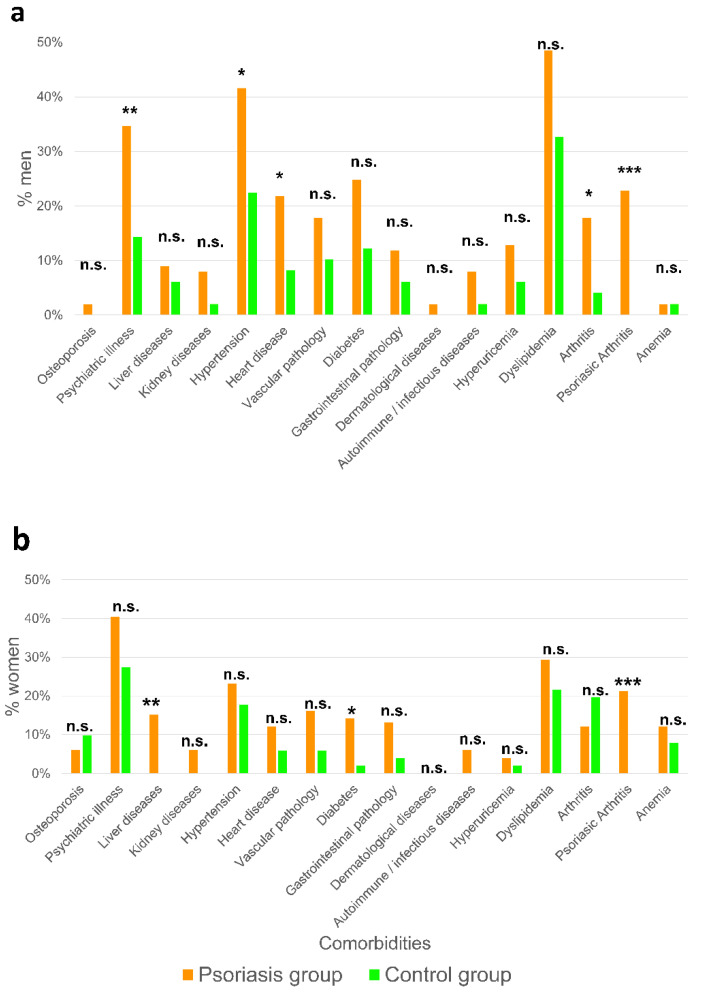
Men have more comorbidities associated to psoriasis than women. Percentage of each comorbidity in psoriasis and control groups in men (**a**) and women (**b**). *p* Values were calculated using a Chi-squared test. * *p* < 0.05; ** *p* < 0.01; *** *p* < 0.001; n.s., non-significant.

**Figure 3 ijms-23-09508-f003:**
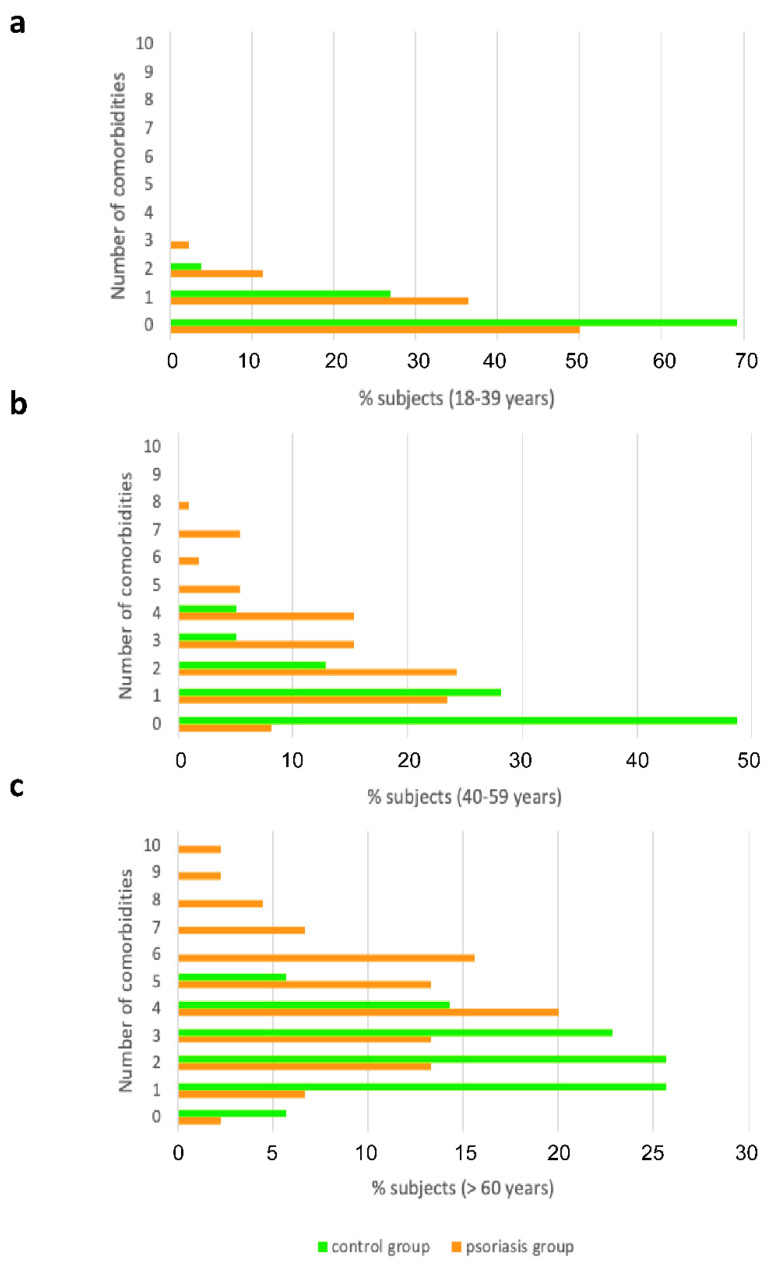
Number of comorbidities in psoriasis and control groups according to their chronological age. Number of comorbidities presented in psoriasis and control groups in the different age groups: 18–39 (**a**), 40–59 (**b**) and >60 (**c**) years old.

**Figure 4 ijms-23-09508-f004:**
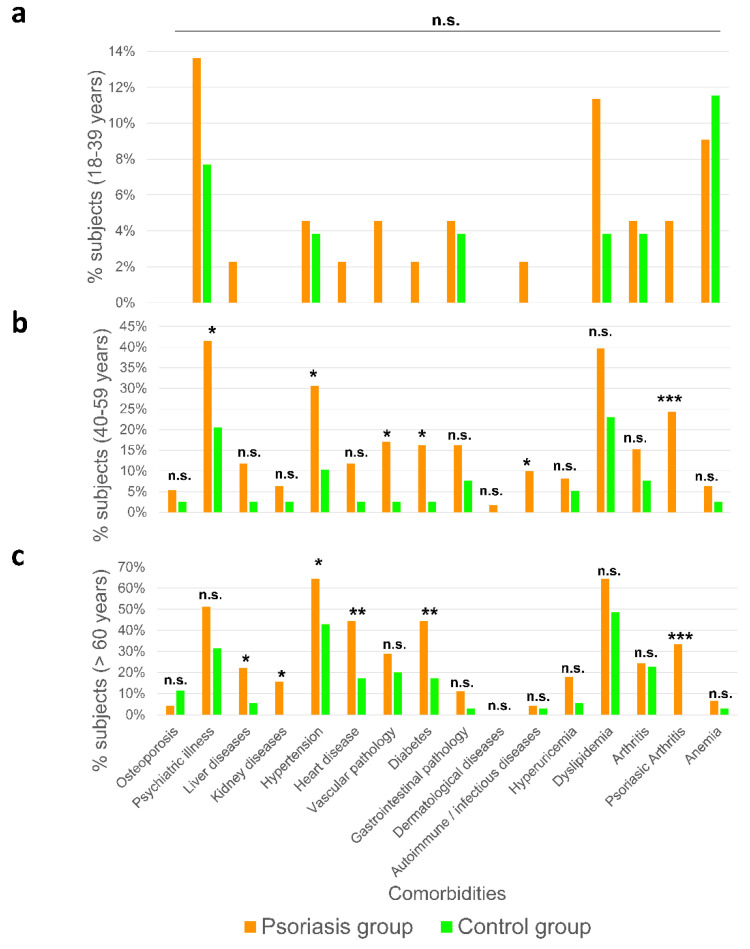
Number of comorbidities in psoriasis and control groups according to their chronological age. Incidence of the different comorbidities of psoriasis and control group subjects of 18–39 (**a**), 40–59 (**b**) and >60 (**c**) years old. *p*-Values were calculated using a Chi-squared test. * *p* < 0.05; ** *p* < 0.01; *** *p* < 0.001; n.s., non-significant.

**Figure 5 ijms-23-09508-f005:**
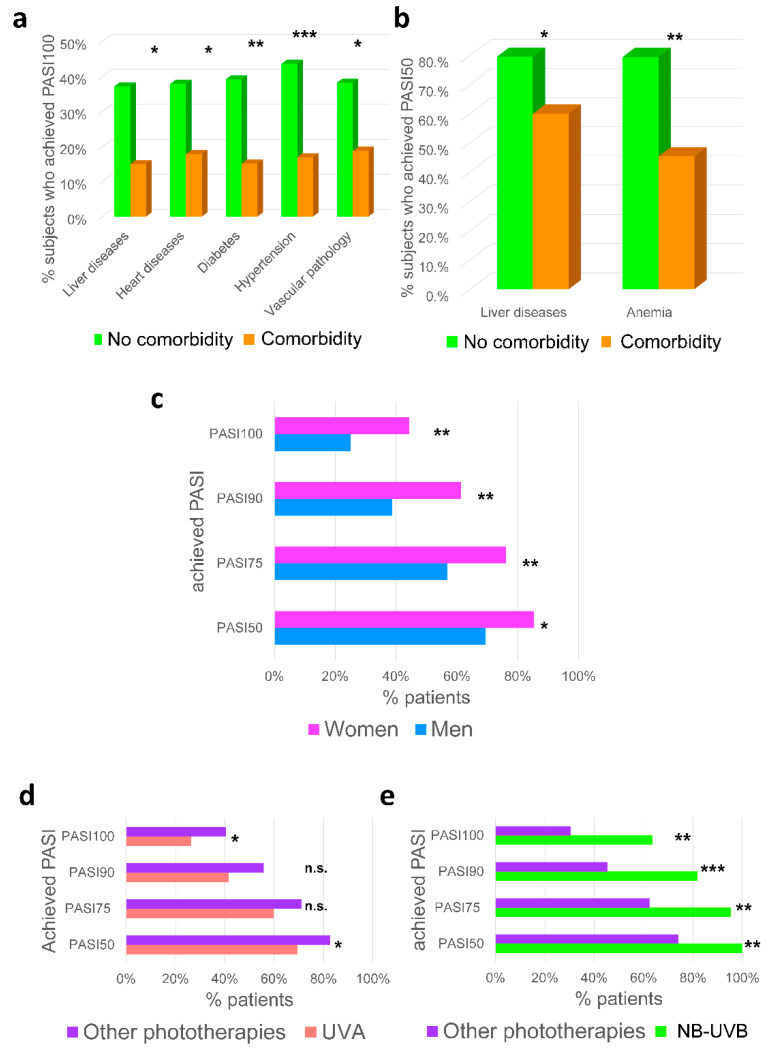
NB-UVB is the best phototherapy for psoriasis treatment. (**a**) Percentage of patients who achieved complete resolution of the lesions (PASI100); (**b**) percentage of patients who achieved PASI50; (**c**) percentage of patients who achieved each PASI depending on gender; (**d**) percentage of patients who achieved each PASI in response to UVA compared with the rest of phototherapies; and (**e**) percentage of patients who achieved each PASI in response to NB-UVB compared with the rest of phototherapies. *p*-Values were calculated using a Chi-squared test. * *p* < 0.05; ** *p* < 0.01; *** *p* < 0.001; n.s., non-significant.

**Figure 6 ijms-23-09508-f006:**
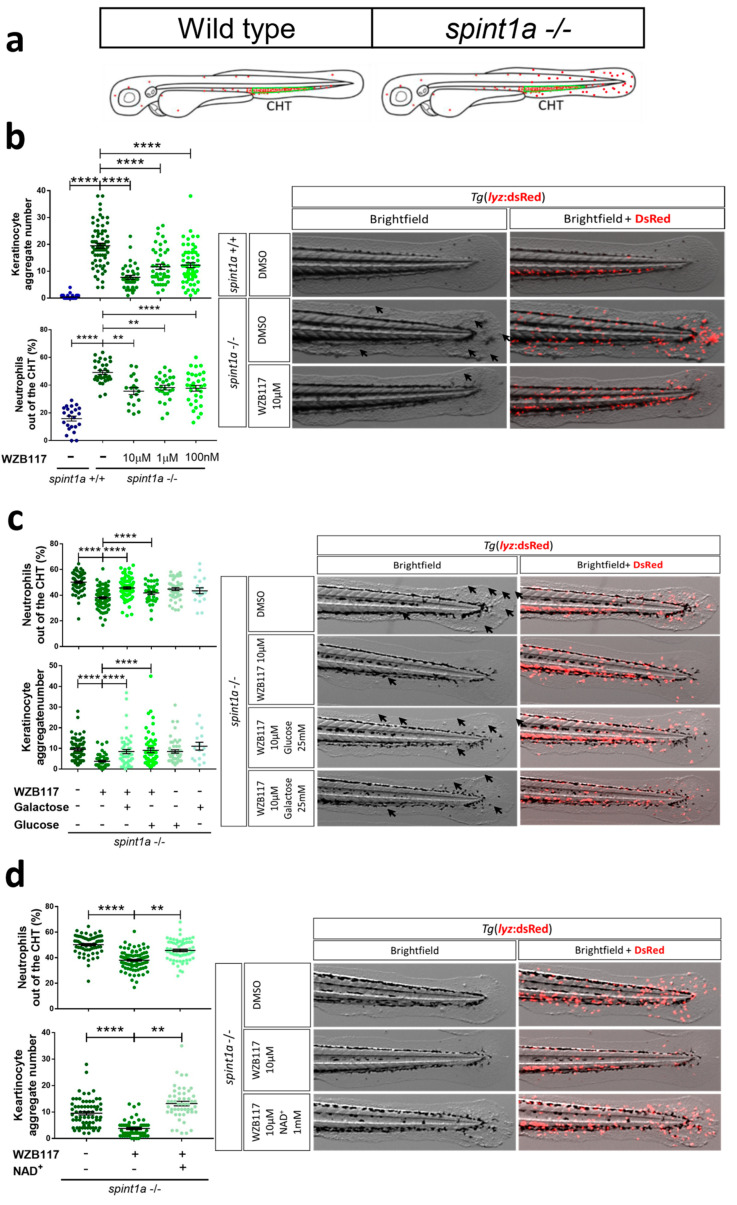
Impact of glucose in zebrafish models of diabetic comorbidity in psoriasis. (**a**) Scheme showing the distribution of neutrophils in wild type and Spint1a-deficient zebrafish larvae. (**b**–**d**) Representative bright field and merge (bright field and red fluorescence) images and quantitation of skin neutrophil infiltration and the number of keratinocyte aggregates (arrows) in 3 dpf larvae treated with 10 µM WZB117 for 48 h in the presence or absence of 25 mM glucose or galactose (**c**), or 1 mM NAD^+^ (**d**). Each dot represents one individual, and the mean ± SEM for each group is also shown. *p*-Values were calculated using 1-way ANOVA and Tukey multiple range test. ns, not significant, ** *p* ≤ 0.01, *** *p* ≤ 0.001, **** *p* ≤ 0.0001.

**Figure 7 ijms-23-09508-f007:**
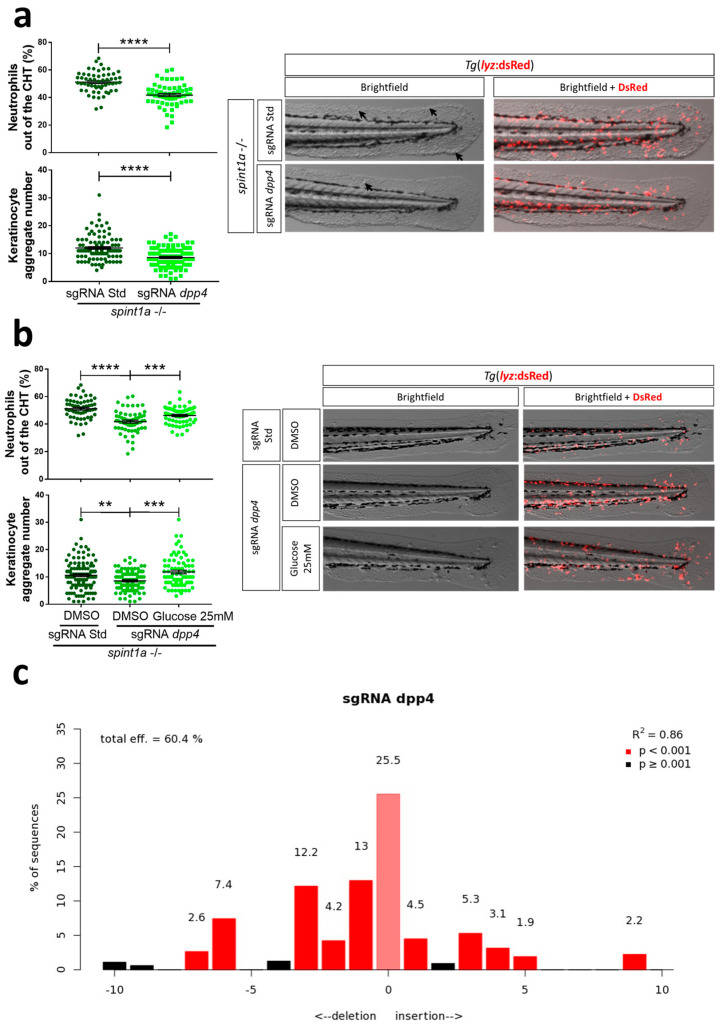
Impact of Dpp4 in zebrafish models of diabetic comorbidity in psoriasis. (**a**–**c**) Representative bright field and merge (bright field and red fluorescence) images and quantitation of skin neutrophil infiltration and the number of keratinocyte aggregates (arrows) in 3 dpf larvae after *dpp4* genetic inhibition using CRISPR/Cas-9 technology in the presence of 25 mM glucose (**b**). (**c**) Analysis of genome editing efficiency in larvae injected with control or *dpp4* crRNA/Cas-9 complexes and quantification rate of NHEJ-mediated repair showing all INDELs (https://tide.nki.nl/). Each dot represents one individual, and the mean ± SEM for each group is also shown. *p*-Values were calculated using 1-way ANOVA and Tukey multiple range test. ns, not significant, ** *p* ≤ 0.01, *** *p* ≤ 0.001, **** *p* ≤ 0.0001.

**Figure 8 ijms-23-09508-f008:**
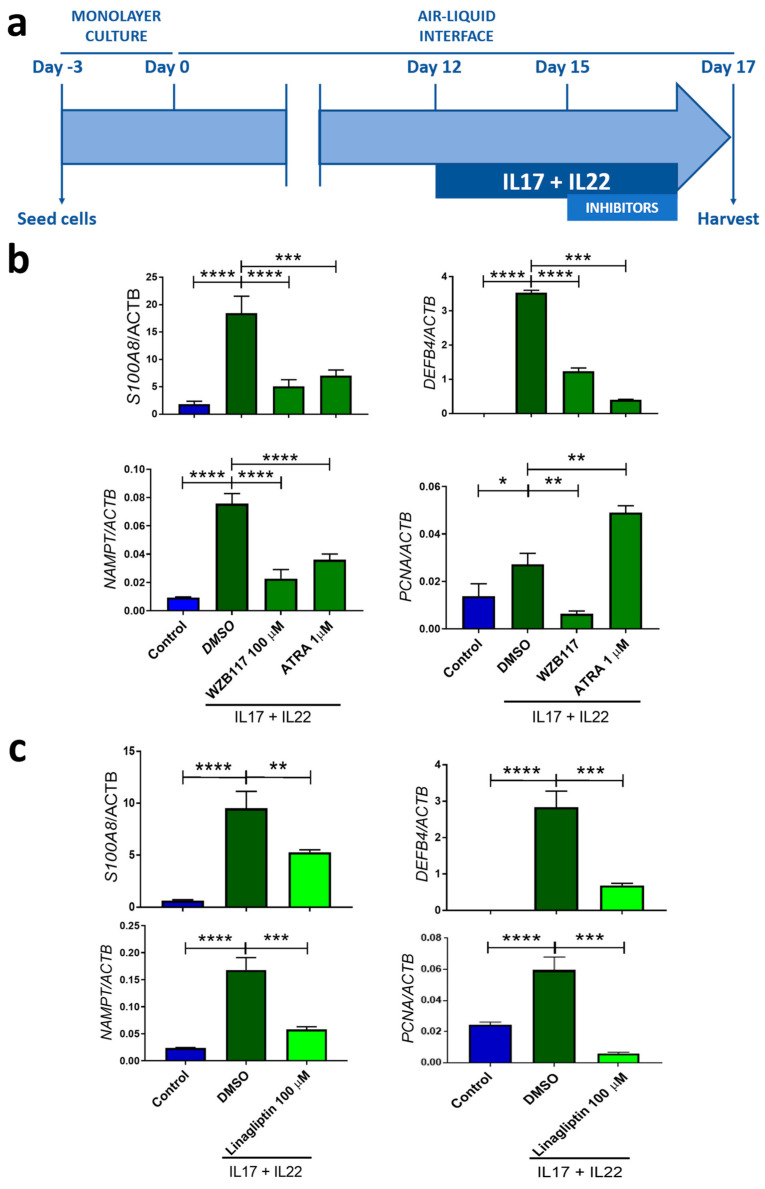
Impact of glucose metabolism in organotypic 3D skin model of human psoriasis. (**a**) Experimental design. (**b**,**c**) The transcript levels of genes encoding pro-inflammatory (S100A8, DEFB4 and NAMPT) and proliferation (PCNA) biomarkers were determined by RT-qPCR after inhibition of GLUT1 with WBZ117 (**b**) or DPP4 with linagliptin (**c**). Treatment with ATRA was used as a positive control. *p*-Values were calculated using 1-way ANOVA and Tukey multiple range test. * *p* ≤ 0.05, ** *p* ≤ 0.01, *** *p* ≤ 0.001, **** *p* ≤ 0.0001.

## Data Availability

All data generated or analyzed during this study are included in this article and its supplementary material files. Further enquiries can be directed to the corresponding authors.

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
