# Peer review of "Impact of Comorbidities of Patients with Psoriasis on Phototherapy Responses"

_ijms, 2022, doi:10.3390/ijms23179508_

Round 1
Reviewer 1 Report
Work on "Impact of comorbidities of patients with psoriasison phototherapy responses" by Fatás-Lalana et al. is interesting and raises an important issue, but requires some significant changes to facilitate readability and reception by readers:
- there is no clearly defined goal of the work;
- I would like to ask you to separate the result part from the discussion section, as it hinders the reception of your research;
- figure 1 is very unreadable, please improve its resolution and use a black font (especially on scales), which will additionally increase its readability. What's more, in Chart 1A there are numerical data (percentages) written as 10,00%, I am asking you to use dotted formatting (i.e. 10.00%). I also propose to increase the font size in both graphs, or maybe use one legend above both graphs, because on both panels psoriasis and control group are marked with the same colors. Below the legend in panel 1B there is a value of p = 0.004 is it necessary here ???
- Figure 2 requires a resolution change. Black font should be used (especially on scales), which will additionally increase its legibility. I also propose to increase the font size in both graphs, or maybe use one legend above both graphs, because on both panels psoriasis and control group are marked with the same colors. The p values on the charts are unreadable, please change them.
- figure 3 is completely illegible and a change of its format should be considered. It presents too much information that is simply lost in this figure and is unreadable. Please, divide it into 2 separate figures and change their resolution, fonts (color and size) in all panels, because in their current form they are very bad. Panel 1 D has two legends, please follow one legend for all panels in this figure. The p values on the charts are unreadable, please change them.
-figure 4 requires rewriting and resolution change. The panels come in different sizes and are completely unreadable. Black font should be used (especially on scales and text), which will additionally increase its readability. There is a number 4 in the chart which is irrelevant and I would like to ask you to remove it. Please consider a different arrangement of the panels in the figure (maybe in 2 columns, thanks to which the graphs will be larger and, hopefully, clearer after changing the font size and arrangement);
- the same applies to figure 5, it is completely to be changed and redrafted, there is absolutely nothing visible on it and these are the key results of the research;
- figure 6 the same, please follow the previous comments. Please also reduce panel 6A, which will allow the remaining panels in this figure to be enlarged and will increase its readability.
- please add a conclusion section
- please add information about additional materials that are cited in the main text in accordance with the editorial requirements in the form.
Despite the very interesting research results, the way they are presented leaves a lot to be desired and has a negative impact on the reception and readability of the work. Please unify the style of data presentation in the figures so that they are legible, in higher resolution and consistent with the text. The lack of a classic division of the text makes the reception of the work more difficult, not to mention the lack of the purpose of the work and the conclusions from the analyzes carried out.
Author Response
Work on "Impact of comorbidities of patients with psoriasison phototherapy responses" by Fatás-Lalana et al. is interesting and raises an important issue, but requires some significant changes to facilitate readability and reception by readers:
- there is no clearly defined goal of the work;
The aim of the study has now been indicated in the last paragraph of the Introduction.
- I would like to ask you to separate the result part from the discussion section, as it hinders the reception of your research;
Thanks for this suggestion but we would like to leave results and discussion together. However, we have added a Conclusions section to better highlight the relevance of our study.
- figure 1 is very unreadable, please improve its resolution and use a black font (especially on scales), which will additionally increase its readability. What's more, in Chart 1A there are numerical data (percentages) written as 10,00%, I am asking you to use dotted formatting (i.e. 10.00%). I also propose to increase the font size in both graphs, or maybe use one legend above both graphs, because on both panels psoriasis and control group are marked with the same colors. Below the legend in panel 1B there is a value of p = 0.004 is it necessary here ???
- Figure 2 requires a resolution change. Black font should be used (especially on scales), which will additionally increase its legibility. I also propose to increase the font size in both graphs, or maybe use one legend above both graphs, because on both panels psoriasis and control group are marked with the same colors. The p values on the charts are unreadable, please change them.
- figure 3 is completely illegible and a change of its format should be considered. It presents too much information that is simply lost in this figure and is unreadable. Please, divide it into 2 separate figures and change their resolution, fonts (color and size) in all panels, because in their current form they are very bad. Panel 1 D has two legends, please follow one legend for all panels in this figure. The p values on the charts are unreadable, please change them.
-figure 4 requires rewriting and resolution change. The panels come in different sizes and are completely unreadable. Black font should be used (especially on scales and text), which will additionally increase its readability. There is a number 4 in the chart which is irrelevant and I would like to ask you to remove it. Please consider a different arrangement of the panels in the figure (maybe in 2 columns, thanks to which the graphs will be larger and, hopefully, clearer after changing the font size and arrangement);
- the same applies to figure 5, it is completely to be changed and redrafted, there is absolutely nothing visible on it and these are the key results of the research;
- figure 6 the same, please follow the previous comments. Please also reduce panel 6A, which will allow the remaining panels in this figure to be enlarged and will increase its readability.
We have carefully revised Figures 1-6 and followed your suggestions. We have also split Fig. 3 and 5 into 2 figures in order to increase graphs and font sizes.
- please add a conclusion section
We have added a conclusion section.
- please add information about additional materials that are cited in the main text in accordance with the editorial requirements in the form.
We are not sure we have to add here. Sorry.
Despite the very interesting research results, the way they are presented leaves a lot to be desired and has a negative impact on the reception and readability of the work. Please unify the style of data presentation in the figures so that they are legible, in higher resolution and consistent with the text. The lack of a classic division of the text makes the reception of the work more difficult, not to mention the lack of the purpose of the work and the conclusions from the analyzes carried out.
Please, see our comments above. Thank you very much for your help to increase the visibility of our work.

Reviewer 2 Report
Paper entitled "Impact of comorbidities of patients with psoriasis on phototherapy responses" deals with contemporary problem of psoriasis treatment and possible resons for treatment failure. Paper is well written with proper methods chosen, and results clearly presented. The authors have thorouglhy examined the problem, and provided some new data that could potentially influence future psoriasis therapy. Conclusions in the paper are supported by the results. After spell checking, I reccomend for the paper to be published.
Author Response
We are pleased with the reviewer's comments on our article
Round 2
Reviewer 1 Report
The work has been improved, but still requires editorial changes. Please justify the main text and format the text as required. According to the journal format, all sections and subsections should be numbered and the conclusion should appear after the materials and methods.
Author Response
We have formmatted the manuscript as requested